# Pip5k1γ regulates axon formation by limiting Rap1 activity

Danila Di Meo[1,2] , Trisha Kundu[1,2], Priyadarshini Ravindran[1] , Bhavin Shah[1], Andreas W Püschel[1,2]

During their differentiation, neurons establish a highly polarized morphology by forming axons and dendrites. Cortical and hippo-campal neurons initially extend several short neurites that all have the potential to become an axon. One of these neurites is then selected as the axon by a combination of positive and negative feedback signals that promote axon formation and prevent the remaining neurites from developing into axons. Here, we show that Pip5k1γ is required for the formation of a single axon as a negative feedback signal that regulates C3G and Rap1 through the generation of phosphatidylinositol-4,5-bisphosphate ($PI(4,5)P_2$). Impairing the function of Pip5k1γ results in a hyper-activation of the Fyn/C3G/Rap1 pathway, which induces the formation of supernumerary axons. Application of a hyper-osmotic shock to modulate membrane tension has a similar effect, increasing Rap1 activity and inducing the formation of supernumerary axons. In both cases, the induction of supernumerary axons can be reverted by expressing constitutively active Pip5k. Our results show that $PI(4,5)P_2$-dependent membrane properties limit the activity of C3G and Rap1 to ensure the extension of a single axon.

## Introduction

Neurons are characterized by their highly polarized morphology with molecularly and functionally distinct axons and dendrites. During their differentiation, they first extend several neurites that all have the potential to become the axon (stage 2) (Schelski & Bradke, 2017). This initial symmetry is broken when one of the neurites begins to extend rapidly as the axon (stage 3), whereas the other neurites do not grow until they differentiate into dendrites (stage 4) (Bradke & Dotti, 2000). This neurite has to extend beyond a minimal length to acquire axonal properties (Goslin & Banker, 1989; Toriyama et al, 2010; Yamamoto et al, 2012). The selection of a single neurite as the axon is controlled by positive and negative feedback signals that promote axon formation and prevent the remaining neu-rites from developing into axons, respectively (Arimura & Kaibuchi, 2007; Barnes & Polleux, 2009; Shelly et al, 2010; Cheng & Poo, 2012; Schelski & Bradke, 2017; Takano et al, 2017, 2019; Wang et al, 2018). Many factors have

been identified that stimulate axon extension, but the molecular nature of the negative feedback signals is not well understood.

GTPases play important roles during axon formation by regu-lating cytoskeletal organization, intracellular trafficking, and plasma membrane expansion (Arimura & Kaibuchi, 2007; Cheng & Poo, 2012; Shah & Püschel, 2014; Schelski & Bradke, 2017; Quiroga et al, 2018). The Rho GTPase Cdc42 and its guanine nucleotide exchange factor (GEF) Arhgef7 are required for axon formation by promoting the exocytosis of specialized vesicles to expand the plasma membrane (López Tobón et al, 2018; Ravindran & Püschel, 2023). Another important pathway for the establishment of neuronal polarity depends on the GEF C3G (Rapgef1) and its substrate, the closely related GTPases Rap1A and Rap1B (Franco et al, 2011; Jossin & Cooper, 2011; Shah & Püschel, 2016; Shah et al, 2017). C3G is phosphorylated at Tyr515 by Src family kinases (SFKs) such as Fyn or Lyn, which stimulates its activity (Ichiba et al, 1999; Ballif et al, 2004; Carabias et al, 2020). The inactivation of C3G or Rap1 GTPases impairs axon formation during cortical and hippocampal development both in vivo and in cultured neurons (Shah et al, 2016, 2017; Wang et al, 2018). Conversely, the inactivation of a GTPase-activating protein as a negative regulator of Rap1 induces the formation of supernumerary axons, in-dicating that Rap1 activity has to be limited to ensure the formation of a single axon (Wang et al, 2018). However, the signals that act upstream of the C3G/Rap1 pathway during the establishment of polarity and restrict its activity to the future axon remain poorly understood.

Several reports suggest that alterations in membrane tension are an important regulator of cell polarity and differentiation (Gauthier et al, 2012; Diz-Muñoz et al, 2013; Le Roux et al, 2019; Sitarska & Diz-Muñoz, 2020). The in-plane tension of the membrane and the membrane-proximal actin cortex that is linked to it determine the effective membrane tension (Sitarska & Diz-Muñoz, 2020). The function of many plasma membrane–associated proteins is regu-lated by phosphatidylinositol-4,5-bisphosphate ($PI(4,5)P_2$), the most abundant phosphoinositide in the plasma membrane (Raucher et al, 2000; Rouven Brückner et al, 2015; Dickson & Hille, 2019). It recruits linker proteins that tether the actin cortex to the plasma membrane and, thereby, contributes to membrane tension (Han et al, 2004; Tamada et al, 2004; Wang et al, 2005; Sawada et al, 2006; Diz-Muñoz et al, 2010; Fehon et al, 2010; Sitarska & Diz-Muñoz, 2020). Reduced levels or impaired metabolism of $PI(4,5)P_2$ have been implicated in various diseases including neurological disorders (Mandal, 2020).

[1]Institut für Integrative Zellbiologie und Physiologie, Universität Münster, Münster, Germany  [2]Cells-in-Motion Interfaculty Center, University of Münster, Münster, Germany

Correspondence: apuschel@uni-muenster.de
Danila Di Meo's present address is European Laboratory for Non-linear Spectroscopy (LENS), University of Florence, Sesto Fiorentino, Italy

PI(4,5)P$_2$ is mainly synthesized by type 1 phosphatidylinositol-4-phosphate 5-kinases (Pip5ks) from phosphatidylinositol 4-phosphate (Stephens et al, 1993; Lemmon, 2008; van den Bout & Divecha, 2009; Liu et al, 2016). The three mammalian Pip5k isoforms Pip5k1α, Pip5k1β, and Pip5k1γ are all expressed in the brain with different patterns during neuronal development (Ishihara et al, 1996; Loijens & Anderson, 1996; Di Paolo et al, 2004; Volpicelli-Daley et al, 2010). In the developing brain, the Pip5k1γ isoform plays a major role in the generation of PI(4,5)P$_2$. The embryonic brain from Pip5k1γ knockout mice shows reduced PI(4,5)P$_2$ levels, and mice lacking Pip5k1γ die after birth, but the role of Pip5ks in neuronal polarization and axon formation remains to be characterized.

Here, we show that Pip5ks are required in developing central neurons for the formation of a single axon. Pip5k1γ functions as a negative feedback signal through the generation of PI(4,5)P$_2$ that regulates the C3G/Rap1 signaling pathway. Application of a hyperosmotic shock to modulate membrane tension increases Rap1 activity and induces the formation of supernumerary axons similar to the effect of impairing the function of Pip5k1γ. In both cases, the induction of supernumerary axons can be reverted by the expression of a constitutively active Pip5k enzyme. Our results show that PI(4,5)P$_2$-dependent membrane properties limit the activity of the C3G/Rap1 pathway to ensure the extension of a single axon.

# Results

## Inactivation of Pip5ks results in the extension of supernumerary axons

Staining with specific antibodies revealed that all Pip5k1 isoforms are expressed in cultured hippocampal neurons before axon formation at 2 d in vitro (d.i.v.) or after polarization at 3 d.i.v. (Fig S1A–C). They are detectable in the somato-dendritic and axonal compartments with only minor differences in their distribution. To explore their function during neuronal polarization, hippocampal neurons from E18 rat embryos were transfected at 0 d.i.v. with knockdown vectors targeting specifically Pip5k1α, Pip5k1β or, Pip5k1γ (Fig S2A–D). Neurons were cultured on poly-L-ornithine, which does not activate integrins, and axon formation was analyzed at 3 d.i.v. (Fig 1A) when neurons have become polarized. Axons and undifferentiated dendrites (called minor neurites) were identified by staining with antibodies for axonal (acetylated tubulin) and dendritic (Map2) markers (Witte & Bradke, 2008; Di Meo et al, 2021; Ravindran & Püschel, 2023). Although most hippocampal neurons extended a single axon in controls (75% ± 2%), many developed supernumerary axons after the knockdown of Pip5k1α (62% ± 1%), Pip5k1β (64% ± 0.4%), or Pip5k1γ (69% ± 1%) (Fig 1B). This phenotype was rescued by the co-expression of RNAi-resistant constructs for the individual Pip5ks, confirming the specificity of the knockdown phenotype (Fig S2A, B, and E). The length of the axons extended after the knockdown of Pip5ks was comparable to controls (Fig S2F). The double knockdown of two different Pip5k isoforms did not further increase the percentage of neurons with supernumerary axons compared with the knockdown of a single isoform. The same

defect in axon formation was also observed when any of the Pip5k isoforms were suppressed in cortical neurons (Fig S2G).

To verify that the knockdown of Pip5ks results in reduced PI(4,5)P$_2$ levels, the PI(4,5)P$_2$ sensor NES-GFP-PLCδ1-PH was expressed together with the knockdown constructs. Quantitative analysis of the fluorescence signals for GFP revealed a significant reduction in the overall level of PI(4,5)P$_2$ after suppression of any of the three Pip5k isoforms (Fig 1C and D).

To exclude that Pip5ks act through a non-catalytic mechanism, we tested the effect of pharmacological inhibitors specific for Pip5k1α (ISA-2011B, 20 mM) or Pip5k1γ (UNC3230, 50 nm). Treatment of neurons with ISA-2011B or UNC3230 at 0 d.i.v. for 72 h resulted in the formation of supernumerary axons (Fig S3A and B; control: 17% ± 1%; ISA-2011B: 48% ± 2%; UNC3230: 65% ± 5%). The combination of both inhibitors led to a similar increase in the percentage of neurons with supernumerary axons, without an apparent additive effect. The phenotype of combining the knockdown of Pip5k1β (for which there are no specific inhibitors commercially available) and the treatment with ISA-2011B and UNC3230 could not be analyzed because neurons did not survive, consistent with a previous analysis that revealed an essential function of Pip5ks for the embryonic development of neurons (Volpicelli-Daley et al, 2010).

## Constitutively active Pip5k1α and Pip5k1β rescue the knockdown of Pip5k1γ

The comparable defects in neuronal polarization after the knockdown of each Pip5k isoform suggest that they do not act redundantly and the function of all three isoforms is required for normal neuronal polarity. To determine the extent to which the different isoforms can compensate for each other, we tested whether the expression of Pip5ks can rescue the knockdown of Pip5k1γ. The overexpression of wild-type Pip5k1α or Pip5k1β did not restore a normal neuronal morphology after suppression of Pip5k1γ, unlike the expression of RNAi-resistant Pip5k1γ (Fig S3C). However, the expression of constitutively active membrane-targeted Pip5k1α or Pip5k1β (Pip5k1αCA or Pip5k1βCA) was able to rescue the loss of Pip5k1γ and restore normal polarity with a single axon (Fig 2A and B). Notably, the expression of Pip5k1αCA or Pip5k1βCA itself did not affect neuronal polarity but strongly reduced axon length in both control and Pip5k1γ knockdown neurons (Fig 2C; control: 322 ± 8 μm; Pip5k1αCA: 167 ± 9 μm; Pip5k1βCA: 193 ± 10 μm; Pip5k1γ RNAi: 317 ± 12 μm; Pip5k1γ RNAi + Pip5k1αCA: 199 ± 13 μm; Pip5k1γ RNAi + Pip5k1βCA: 200 ± 10 μm). Therefore, the constitutive recruitment of Pip5k1α or Pip5k1β to the plasma membrane is sufficient to support the production of PI(4,5)P$_2$ that blocks the formation of supernumerary axons. Taken together, these results indicate that Pip5ks are essential for the establishment of normal neuronal polarity by preventing the formation of supernumerary axons and ensuring the extension of a single axon.

## Pip5k1γ regulates the Fyn/C3G/Rap1 pathway

Rap1 GTPases and their activator C3G are pivotal components of a signaling pathway that promotes axon formation (Shah & Püschel,

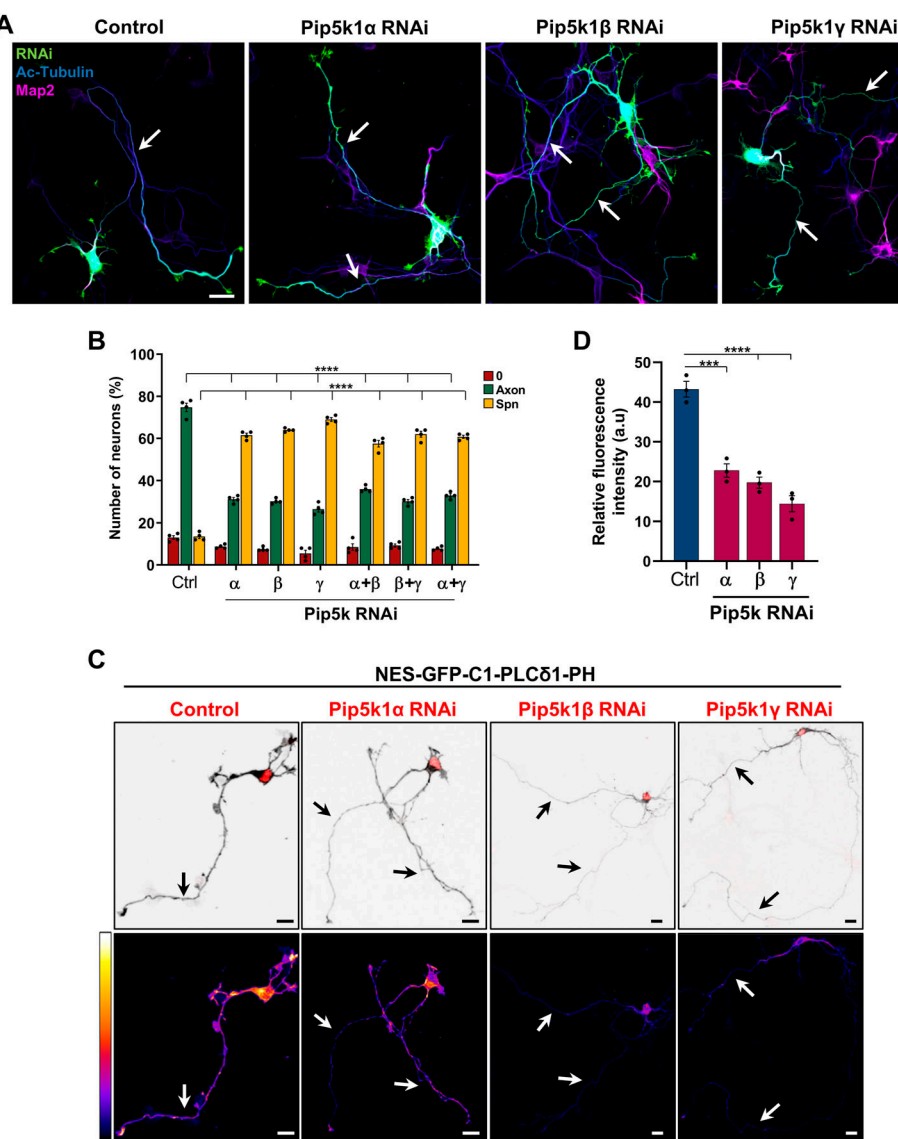

**Figure 1. Knockdown of Pip5ks induces the formation of supernumerary axons.**
**(A)** Hippocampal neurons were transfected with control or knockdown vectors directed against Pip5k1α, Pip5k1β, or Pip5k1γ expressing GFP (RNAi, green) and stained with anti-acetylated tubulin (blue, axonal marker) and anti-Map2 (magenta) antibodies. **(A, B)** Percentage of unpolarized neurons without an axon (0), polarized neurons with a single axon (Axon), and neurons with supernumerary axons (Spn) is shown (n = 4 independent experiments that are biological replicates). **(C)** Hippocampal neurons were transfected with a vector for NES-GFP-PH-PLCδ1 and control or knockdown vectors for Pip5k1α, Pip5k1β, or Pip5k1γ (red). Lower panels display the relative fluorescence intensity (pseudocolor scale: bright yellow and black represent the maximum and minimum intensity, respectively). **(C, D)** Relative fluorescence intensity of NES-GFP-PH-PLCδ1 is quantified for (C) (n = 3 independent experiments that are biological replicates). White/black arrows mark axons. Scale bars: 20 μm. Values are means ± SEM. **(B, D)**: one-way ANOVA.

2016; Shah et al, 2017). The activation of C3G depends on its phosphorylation at Tyr515 by SFKs such as Fyn (Carabias et al, 2020). The overexpression of constitutively active Rap1BV12 induces the formation of supernumerary axons in hippocampal neurons similar to the effect of suppressing Pip5ks (Wang et al, 2018). The formation of multiple axons upon suppression of Pip5ks could, therefore, indicate an increased activity of Rap1. To address this question, we examined whether the loss of Pip5k1γ alters the activity of Fyn, C3G, and Rap1. Phosphorylation of a specific tyrosine residue in the conserved activation loop of the kinase domain is indicative of SFK activation (Roskoski, 2004, 2005; Matrone et al, 2020). Activated SFKs were immunoprecipitated with a phospho-SFK-Tyr416 antibody, and the amount of active Fyn was quantified with an anti-Fyn antibody (Du et al, 2020). Inhibition of Pip5k1γ activity at 0 d.i.v. for 72 h resulted in a strong increase in the levels of Fyn phosphorylated at Tyr416, indicative of its activation (Fig 3A and B). The

same effect was also observed when the inhibitor was added at 2 d.i.v. for 5 h, and Fyn phosphorylation was analyzed immediately after the treatment. To determine whether the increased activity of Fyn is accompanied by activation of C3G and Rap1, the amount of endogenous C3G phosphorylated at Tyr515 and active GTP-bound Rap1 was quantified. Inhibition of Pip5k1γ for 72 h resulted in increased levels of phosphorylated C3G, indicative of its activation. Moreover, a pull-down assay with bacterially expressed GST-RBD (Ras/Rap-binding domain) (Ren et al, 1999) revealed a strong increase in the amount of active Rap1 (Fig 3C–E). A similar change in C3G phosphorylation and the amount of active Rap1 was also observed when Pip5k1γ was inhibited at 2 d.i.v. for 5 h, and neurons were directly analyzed after treatment. However, when neurons were treated with UNC3230 at 1 d.i.v. for 5 h and cultured until 3 d.i.v., the levels of active Rap1 and phospho-Tyr515-C3G were not different from controls (Fig S3E and F) although this treatment

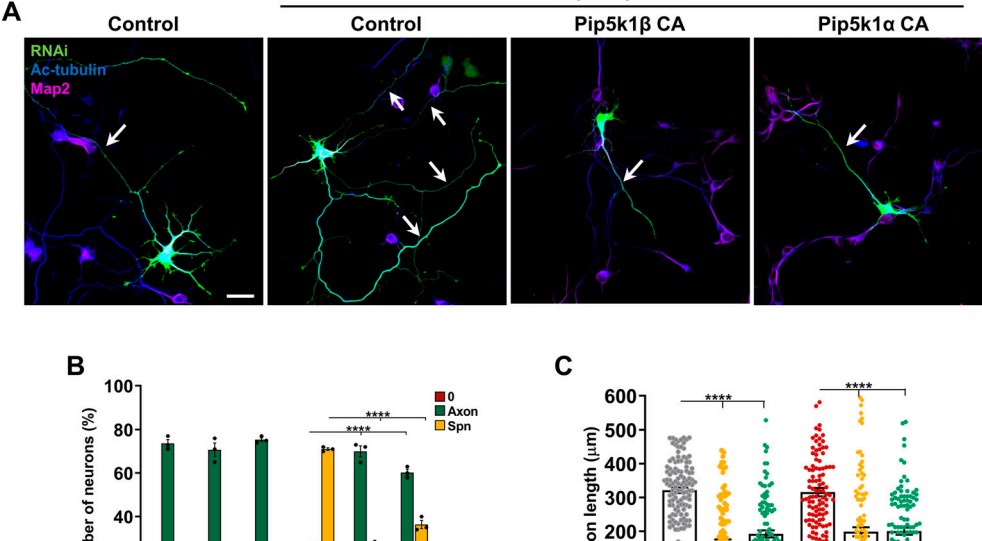

**Figure 2. Expression of constitutively active Pip5ks reverses the formation of supernumerary axons.**
**(A)** Hippocampal neurons were transfected with vectors for constitutively active Pip5k1α or Pip5k1β (Pip5k1αCA and Pip5k1βCA) and a control or Pip5k1γ knockdown vector and stained with anti-acetylated tubulin (blue) and anti-Map2 (magenta) antibodies. White arrows indicate axons. **(A, B)** Percentage of unpolarized neurons without an axon (0), polarized neurons with a single axon (Axon), and neurons with supernumerary axons (Spn) is shown for (A) (n = 3 independent experiments that are biological replicates). **(A, C)** Length of single or supernumerary axons is shown for (A) (n = 3 independent experiments that are biological replicates). Scale bars: 20 $\mu m$. Values are means ± SEM. **(B, C)**: one-way ANOVA.

induces the extension of supernumerary axons (Fig S3D). This result indicates that the increase in active Rap1 levels after inhibition of Pip5k1γ is not merely a consequence of the formation of supernumerary axons. Thus, a transient inhibition of Pip5k1γ in unpolarized neurons is sufficient to stimulate the activation of Rap1 and induce the formation of supernumerary axons, but a sustained Pip5k1γ inhibition is required to maintain increased C3G/Rap1 activity.

### Pip5k1γ restricts the C3G/Rap1-dependent formation of axons

Immunofluorescence analysis of neurons that had not yet fully polarized (2 d.i.v.) revealed that phospho-Tyr515-C3G and active Rap1 are preferentially localized to a single neurite in control neurons (Fig 3F and H). Interestingly, the inhibition of Pip5k1γ for 5 h at 2 d.i.v. leads to a non-polarized distribution of both phospho-Tyr515-C3G and active Rap1 in multiple neurites. The polarization index of phospho-Tyr515-C3G decreased from 3.3 ± 0.2 a.u. in controls to 1.6 ± 0.1 a.u. in treated neurons, and of active Rap1, from 3.6 ± 0.3 a.u. in control neurons to 1.6 ± 0.1 a.u. in treated neurons (Fig 3G and I). These results indicate that the production of PI(4,5)P$_2$ is required to restrict Rap1 function to a single neurite.

To confirm the involvement of C3G and Rap1, neurons were transfected with knockdown vectors for C3G or Rap1 and Pip5k1γ activity was inhibited by treatment with UNC3230 for 72 h (Figs 4A and B and S3G and H). The knockdown of Rap1 or C3G blocked axon formation in controls, as reported before (Schwamborn & Püschel, 2004; Shah & Püschel, 2016; Shah et al, 2017). The formation of supernumerary axons after inhibition of Pip5k1γ was suppressed by the knockdown of either C3G or Rap1 (Fig 4C and D).

These results confirm that the phenotype induced by the loss of Pip5k1γ results from an over-activation of the C3G/Rap1 cascade. Together, these findings show that PI(4,5)P$_2$ produced by Pip5k1γ acts as a signal that restricts the localization and activation of Rap1 to a single neurite by regulating Fyn and C3G activity. Suppression of the Pip5k1γ-mediated negative feedback signal results in activation of the Fyn/C3G/Rap1 pathway in multiple neurites, leading to the extension of supernumerary axons.

### Hyper- and hypo-osmotic shock disrupts neuronal polarization

PI(4,5)P$_2$ regulates plasma membrane homeostasis and tension, which is emerging as an important regulator of cell polarity and differentiation (Diz-Muñoz et al, 2010; Fehon et al, 2010; Sitarska & Diz-Muñoz, 2020; Bergert et al, 2021; De Belly et al, 2023). A well-established method to manipulate membrane tension is the application of hyper- or hypo-osmotic shocks that decrease and increase tension, respectively (Sitarska & Diz-Muñoz, 2020; Roffay et al, 2021; Ho et al, 2023). To test whether changes in membrane tension affect axon formation, we exposed hippocampal neurons to an osmotic shock before neuronal polarity is established and analyzed axon formation at 3 d.i.v. (Fig 5A). A hyper-osmotic shock was induced by replacing the normal culture medium (Neurobasal medium, NBM) with NBM supplemented with 50 mM sucrose or 50 mM sorbitol at 1 d.i.v. for 48 h. Both treatments resulted in the development of supernumerary axons at 3 d.i.v. (Fig 5B and C; supernumerary axons: 17% ± 6% [control], 42% ± 1% [sucrose], 52% ± 4% [sorbitol]). The same phenotype was also observed when a hyper-osmotic shock was applied for 5 h at 1 d.i.v. (Fig 5B and C;

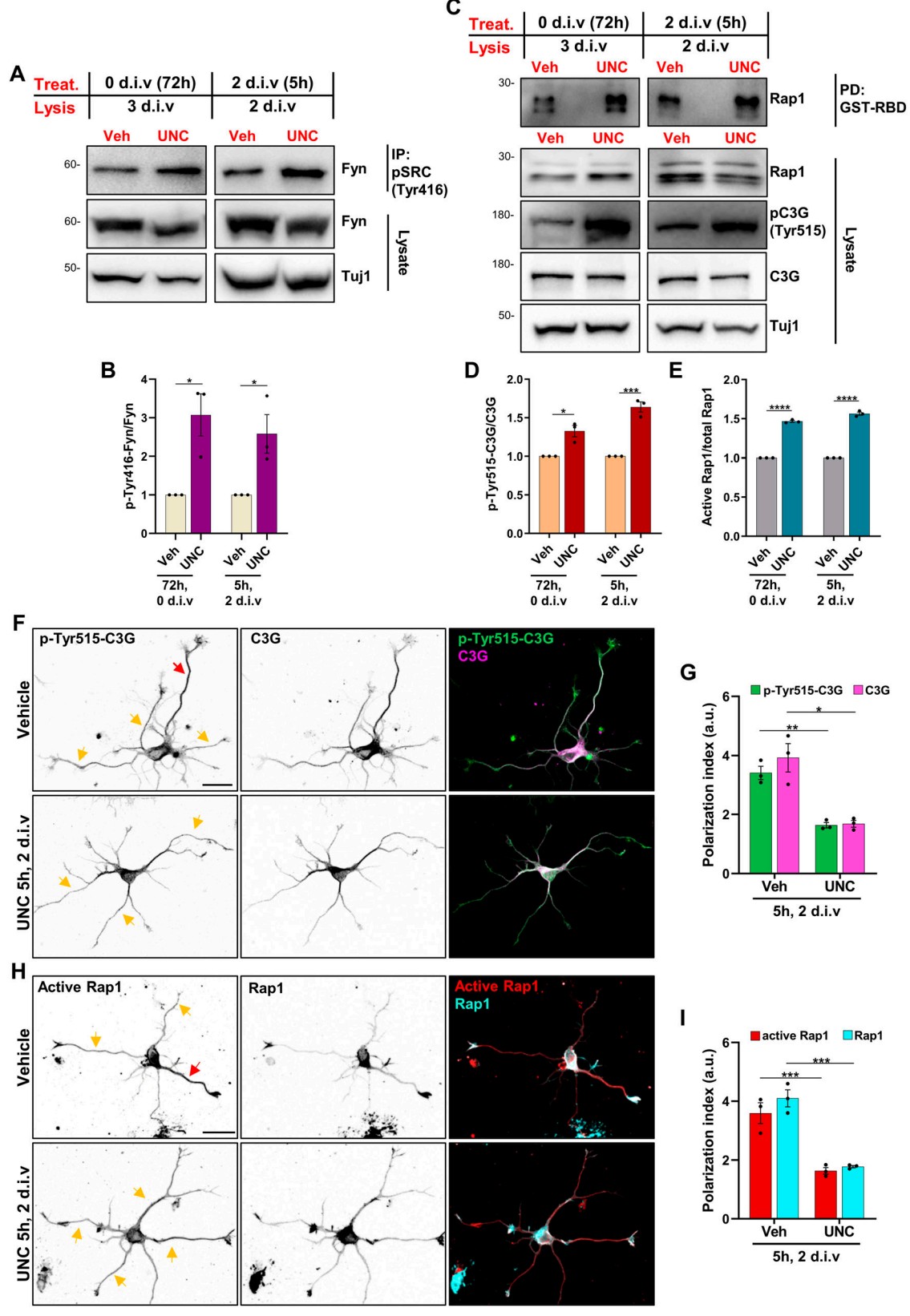

supernumerary axons: 14% ± 2% [control], 50% ± 1% [sucrose], 51% ± 4% [sorbitol]). Conversely, axon formation was impaired when neurons were exposed to a hypo-osmotic shock at 1 d.i.v. for 48 h (NBM diluted to 75% or 50% with water; Fig 5D and E; unpolarized neurons: 17% ± 3% [control], 53% ± 9% [75% NBM], 68% ± 4% [50% NBM]). Axon formation was also affected after a hypo-osmotic shock at 1 d.i.v. for 5 h (Fig 5D and E; unpolarized neurons: 18% ± 4% [control], 38% ± 7% [75% NBM], 52% ± 9% [50% NBM]). An inhibition of axon formation was also observed when neurons were subjected to a hypo-osmotic shock by culturing them from 0 to 1 d.i.v. in a medium with high osmolarity (NBM with 50 mM sucrose or sorbitol) and switching the medium to normal NBM at 1 d.i.v. (Fig 5F and G; unpolarized neurons: 16% ± 2% [control], 55% ± 8% [sucrose], 60% ± 3% [sorbitol]). Thus, the manipulation of membrane properties by an osmotic shock affects axon formation. A hyper-osmotic shock leads to the formation of multiple axons, whereas a hypo-osmotic shock has the opposite effect and blocks the axon formation.

### The effect of a hyper-osmotic shock depends on Pip5k1γ and C3G/Rap1

Alterations in membrane tension can modulate the activity of SFKs (Han et al, 2004; Tamada et al, 2004; Wang et al, 2005; Sawada et al, 2006). To determine whether the induction of supernumerary axons by a hyper-osmotic shock requires the C3G/Rap1 pathway, we first tested whether it affects the activity of Rap1 GTPases. Neurons were subjected to a hyper-osmotic shock for 5 h at 2 d.i.v., and the amount of active Rap1 was directly determined after treatment (Fig 6A and B). Similar to the inhibition of Pip5k1γ, both treatments resulted in a strong increase in the levels of active Rap1. To determine whether the induction of supernumerary axons depends on C3G and Rap1, hippocampal neurons were transfected with knock-down vectors and subjected to a hyper-osmotic shock at 1 d.i.v. for 48 h (Fig 6C and D). The formation of supernumerary axons after a hyper-osmotic shock was suppressed, and most of the neurons extended a single axon after the knockdown of either C3G (Fig 6C and D; supernumerary axons after knockdown: 46% ± 2% [control treatment], 29% ± 1% [sucrose]; 48% ± 2% [control treatment], 29% ± 1% [sorbitol]) or Rap1 (Fig 6E and F; supernumerary axons after knockdown: 40% ± 1% [control treatment], 22% ± 3% [sucrose]; 42% ± 2% [control treatment], 25% ± 2% [sorbitol]). To determine whether the effect of a hyper-osmotic shock is linked to the function of Pip5ks, we tested whether the

simultaneous expression of constitutively active Pip5k1αCA can restore normal polarity. Interestingly, Pip5k1αCA was able to compensate for the effect of a hyper-osmotic shock, allowing most of the neurons to extend a single axon (Fig 6G and H; supernumerary axons: 41% ± 1% [control + sucrose], 29% ± 3% [−αCA + sucrose], 44% ± 3% [control + sorbitol], 28% ± 2% [−αCA + sorbitol]). Axon length was reduced compared with control after a hyper-osmotic shock (Fig S4A), suggesting that it may modulate pathways in addition to Pip5ks (Diz-Muñoz et al, 2016; Loh et al, 2019) Axon length was restored to control levels by the expression of Pip5k1αCA. A hyper-osmotic shock also significantly reduced the level of PI(4,5)$P_2$ in neurons at 3 d.i.v. (Fig S4B and C). Taken together, these results suggest that the reduced levels of PI(4,5)$P_2$ after interfering with Pip5k1γ are linked to a reduction in plasma membrane tension. Both result in the stimulation of the C3G/Rap1 pathway in multiple neurites, leading to the formation of supernumerary axons.

## Discussion

The formation of an axon is orchestrated by several GTPases that promote the polarization of intracellular trafficking, cytoskeletal dynamics, and the expansion of the plasma membrane (Pfenninger, 2009; Kapitein & Hoogenraad, 2015; Schelski & Bradke, 2017; Quiroga et al, 2018). In this study, we show that PI(4,5)$P_2$ produced by Pip5kγ acts as a negative feedback signal in developing neurons that limits the activity of Rap1 GTPases, thereby ensuring the extension of a single axon (Fig 7). Knockdown or inhibition of Pip5kγ results in a de-repression of the C3G/Rap1 activity in neurites that later become dendrites and normally display a low activity of this pathway (Fig 7A). This hyper-activation in several neurites induces the formation of supernumerary axons (Fig 7B).

Changes in the amount of PI(4,5)$P_2$ may also affect the levels of other phosphoinositides, and we cannot exclude the possibility that this also contributes to the polarity defect (Ménager et al, 2004; Raghu et al, 2019). However, the restoration of normal polarity in Pip5k1γ knockdown neurons by active Pip5ks suggests that the production of PI(4,5)$P_2$ is the major factor. The hyper-activation of the C3G/Rap1 pathway upon loss of Pip5k1γ is likely mediated by the activation of Fyn as indicated by its increased phosphorylation at Tyr416 and the increased phosphorylation of its target C3G at Tyr515. Both total and phosphorylated C3G and total and GTP-bound Rap1

---

**Figure 3.  Inhibition of Pip5k1γ activates the Fyn/C3G/Rap1 pathway.**
**(A)** Cortical neurons were treated with vehicle (Veh, DMSO) or UNC3230 (UNC) as indicated. Src family kinases phosphorylated at Tyr416 were immunoprecipitated and analyzed by Western blot with an anti-Fyn antibody as indicated. Detection of Tuj1 confirmed the analysis of comparable amounts of lysate. **(A, B)** Normalized ratio of phospho-Tyr416-Fyn to total Fyn is shown (n = 3 independent experiments that are biological replicates). **(C)** Cortical neurons were treated as in (A), and the amount of active GTP-bound Rap1 was quantified by a pull-down assay using bacterially expressed GST-RBD (Ras/Rap-binding domain) and analyzed by Western blot with the indicated antibodies. The levels of phospho-Tyr515-C3G and total C3G were analyzed from lysates of cultured neurons. Detection of Tuj1 confirmed the analysis of comparable amounts of lysate. **(C, D, E)** Normalized ratio of active to total Rap1 (D) and phospho-Tyr515-C3G to total C3G (E) is shown (n = 3 independent experiments that are biological replicates). The molecular weight is indicated in kD. **(F, H)** Hippocampal neurons were treated with vehicle (DMSO) or UNC3230 for 5 h at 2 d.i.v. and stained with anti-phospho-Tyr515-C3G (green) and anti-C3G (magenta) (F) or with anti-GTP-Rap1 (red) and anti-Rap1 (cyan, total Rap1) (G) antibodies. Red arrows mark the neurite with the strongest signal, whereas yellow arrows mark the other neurites. **(F, G, H, I)** Polarization index was calculated for (F, H) as the fluorescence intensity (a.u.) of the brightest neurite divided by the average fluorescence intensity (a.u.) of the other minor neurites of the same cell. Scale bar: 20 μm. Values are means ± SEM. **(B, D, E, G, I)**: unpaired t test.
Source data are available for this figure.

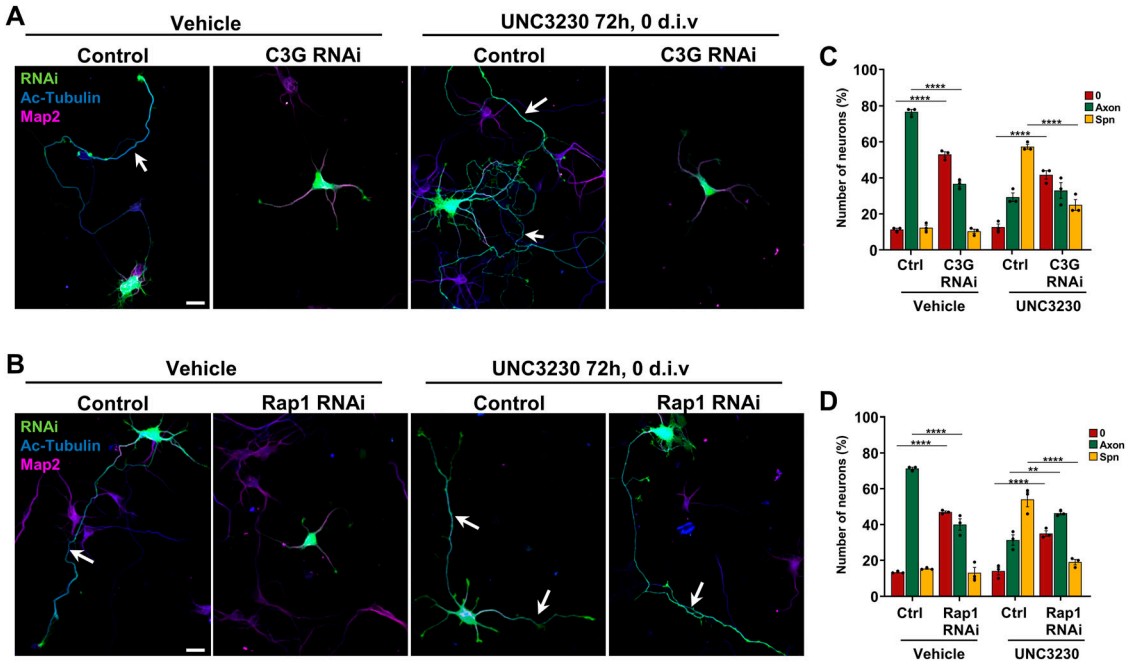

**Figure 4. C3G and Rap1 are required for the formation of supernumerary axons after inhibition of Pip5k1γ.**
**(A, B)** Hippocampal neurons were transfected with control or knockdown vectors (RNAi, green) directed against C3G (A) or Rap1 (B), treated with vehicle (DMSO) or UNC3230 for 72 h, and stained with anti-acetylated tubulin (blue) and anti-Map2 (magenta) antibodies. White arrows mark axons. Scale bars: 20 $\mu$m. **(A, B, C, D)** Percentage of unpolarized neurons without an axon (0), polarized neurons with a single axon (Axon), and neurons with supernumerary axons (Spn) is shown for (A, B) (n = 3 independent experiments that are biological replicates). Values are means ± SEM. **(C, D)**: one-way ANOVA.

preferentially localize to a single neurite in hippocampal neurons that are not fully polarized. This polarized distribution is lost upon inhibition of Pip5k1γ. Together with their increased activity, this results in the extension of supernumerary axons after inactivation of Pip5kγ because normal polarity was restored by a knockdown of C3G or Rap1. Taken together, our results indicate that Pip5k1γ mediates the restricted activity of the C3G/Rap1 pathway in a single neurite.

The formation of supernumerary axons was also observed upon interfering with Pip5k1α or Pip5k1β. Different Pip5k isoforms are known to play distinct roles in non-neuronal cells (van den Bout & Divecha, 2009). Our results suggest that all three isoforms are required for normal neuronal polarity and do not act redundantly during axon formation. The overexpression of full-length, wild-type Pip5k1α or Pip5k1β did not rescue the knockdown of Pip5k1γ, confirming that one isoform cannot compensate for the reduced expression of another. In contrast, the expression of Pip5k1α or Pip5k1β constructs that are targeted to the plasma membrane and constitutively active rescued the loss of Pip5k1γ and restored normal polarity with a single axon. Pip5ks can form homo- and heterodimers, which influences their localization and activity (Hu et al, 2015; Lacalle et al, 2015; Hansen et al, 2022) and may explain why Pip5ks do not function redundantly during axon formation. However, this possibility remains to be tested directly because the functional properties of these heterodimers have not been characterized in sufficient detail.

PI(4,5)P$_2$ is implicated in the regulation of membrane tension because it recruits several actin-binding proteins including members of the ezrin/radixin/moesin family that link the actin cortex to the plasma membrane and, thereby, contribute to effective membrane tension (Tsukita & Yonemura, 1999; Braunger et al, 2014; Matsumoto et al, 2014; McClatchey, 2014; Rouven Brückner et al, 2015). Membrane tension is an important regulator of cell polarity and differentiation (Diz-Muñoz et al, 2010, 2013; Fehon et al, 2010; Gauthier et al, 2012; Houk et al, 2012; Saha et al, 2018; Le Roux et al, 2019; Sitarska & Diz-Muñoz, 2020; Bergert et al, 2021; De Belly et al, 2023). The effects observed upon applying an osmotic shock as an established method to induce alteration in plasma membrane tension suggest that it plays an important role in the regulation of axon formation (Dai et al, 1998; Alam Shibly et al, 2016).

The increased activity of Rap1 after inhibition of Pip5k1γ or hyper-osmotic shock probably results from a stimulation of SFKs. In non-neuronal cells, mechanical stimulation activates members of the SFKs and Rap1 signaling (Sawada et al, 2001, 2006; Han et al, 2004; Tamada et al, 2004; Wang et al, 2005; Gauthier et al, 2012; Heisenberg & Bellaı, 2013; Le Roux et al, 2019). A mild hyper-osmotic shock increased the amount of active Rap1 and induced the formation of supernumerary axons, similar to the inhibition of Pip5k1γ, whereas a mild hypo-osmotic shock inhibited the development of axons. Importantly, the formation of supernumerary axons after a hyper-osmotic shock was prevented by the knockdown of C3G or Rap1, and by the expression of constitutively active Pip5k1αCA. The ability to revert the induction of supernumerary axons supports a link between the production of PI(4,5)P$_2$ and the effects of manipulating membrane tension. This suggests that reduced levels of PI(4,5)P$_2$ might alter membrane tension, for example, because of decreased coupling to the actin cortex, leading to an increased activity of Fyn and the de-repression of the C3G/Rap1 pathway in multiple neurites.

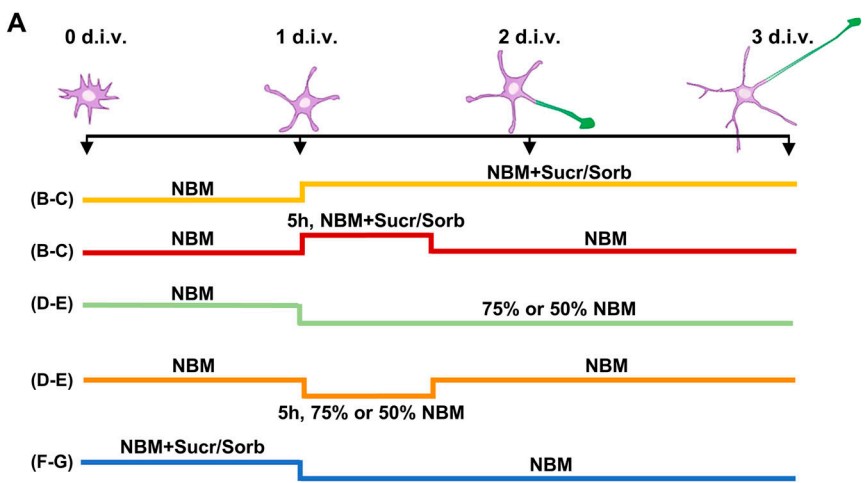

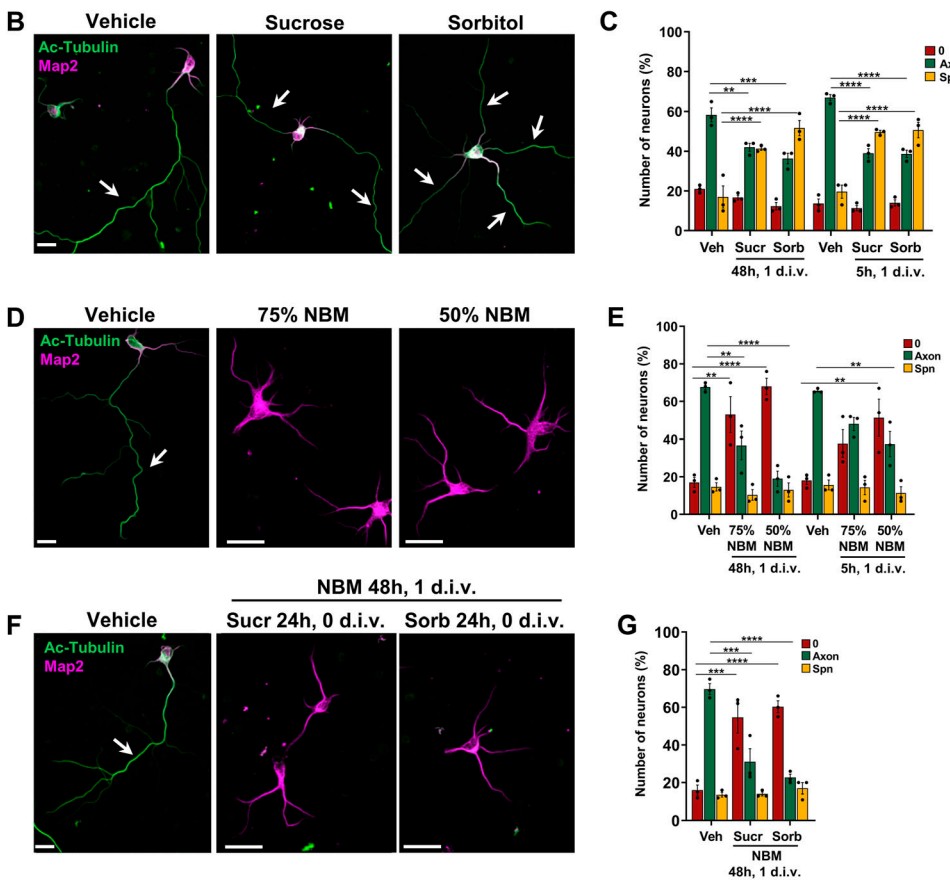

**Figure 5. Osmotic shock disrupts the polarization of neurons.**
**(A)** Schematic representation of the culture conditions used to expose neurons to a hyper- or hypo-osmotic shock. **(B)** Hippocampal neurons were exposed to a hyper-osmotic shock at 1 d.i.v. for 48 or 5 h by replacing the culture medium NBM (vehicle) with NBM supplemented with 50 mM sucrose or sorbitol as indicated. Neurons were stained at 3 d.i.v. with anti-acetylated tubulin (green) and anti-Map2 (magenta) antibodies. **(B, C)** Percentage of unpolarized neurons without an axon (0), polarized neurons with a single axon (Axon), and neurons with supernumerary axons (Spn) is shown for (B) (n = 3 independent experiments that are biological replicates). **(D)** Hippocampal neurons were exposed to a hypo-osmotic shock at 1 d.i.v. for 48 or 5 h by replacing the culture medium NBM (vehicle) with NBM diluted to 75% or 50% NBM with water. Neurons were stained at 3 d.i.v. with anti-acetylated tubulin (green) and anti-Map2 (magenta) antibodies. **(D, E)** Percentage of unpolarized neurons without an axon (0), polarized neurons with a single axon (Axon), and neurons with supernumerary axons (Spn) is shown for (D) (n = 3 independent experiments that are biological replicates). **(F)** Hippocampal neurons were cultured in NBM (vehicle) or 50 mM sucrose- or sorbitol-supplemented NBM as indicated for 24 h and exposed to a hypo-osmotic shock by replacing the medium at 1 d.i.v. with normal NBM. Neurons were stained at 3 d.i.v. with anti-acetylated tubulin (green) and anti-Map2 (magenta) antibodies. **(F, G)** Percentage of unpolarized neurons without an axon (0), polarized neurons with a single axon (Axon), and neurons with supernumerary axons (Spn) is shown for (F) (n = 3 independent experiments that are biological replicates). Scale bars: 20 $\mu$m. Values are means ± SEM. **(C, E, G)**: one-way ANOVA.

It will be interesting to investigate in future experiments whether membrane tension can act as a long-range signal between neurites. It was initially reported that membrane tension is not transmitted readily over long distances (Shi et al, 2018). Recent results indicate that changes in membrane tension can be propagated when the actin cortex is engaged because of its tethering to the plasma membrane (Shi et al, 2018, 2022; Gomis Perez et al, 2022; De Belly et al, 2023). It remains to be investigated whether local changes in membrane properties are transmitted to the other neurites to regulate their extension. Taken together, our results suggest that all Pip5k isoforms are required for the normal establishment of neuronal polarity. In developing neurons, PI(4,5)P$_2$ produced by Pip5k1$\gamma$- and PI(4,5)P$_2$-dependent membrane properties linked to membrane tension limit the activity of the C3G/Rap1 pathway, thereby ensuring the formation of neurons with a single axon.

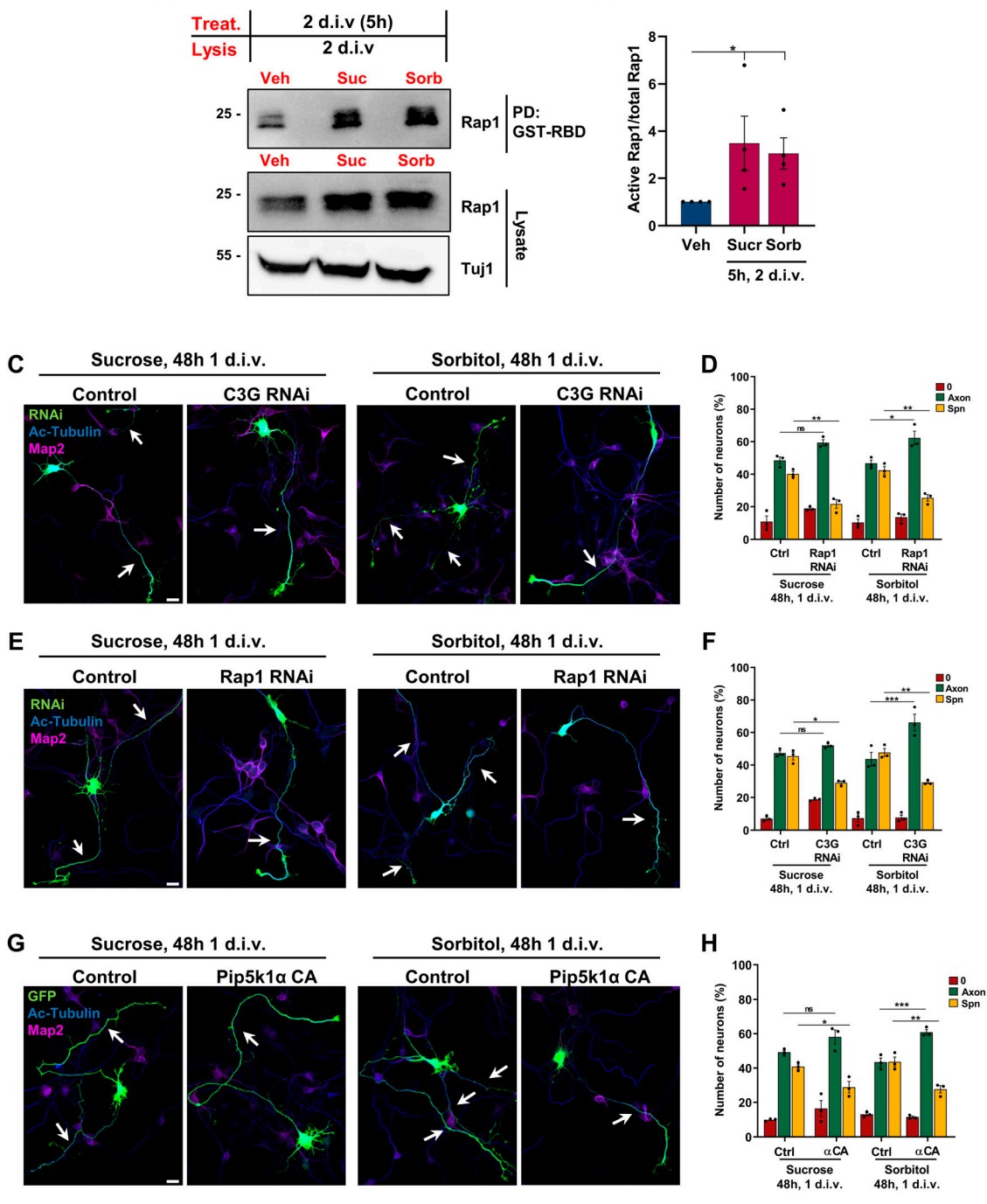

**Figure 6. Hyper-osmotic shock increases Rap1 activity and disrupts neuronal polarization.**
**(A)** Cortical neurons were exposed to a hyper-osmotic shock at 2 d.i.v. for 5 h by replacing the culture medium NBM (Veh) with NBM supplemented with 50 mM sucrose or sorbitol as indicated. The amount of active GTP-bound Rap1 was quantified by pull-down assays using bacterially expressed GST-RBD (Ras/Rap-binding domain) and analyzed by Western blot with the indicated antibodies. Detection of Tuj1 confirmed the analysis of comparable amounts of lysate. The molecular weight is indicated in kD.
**(A, B)** Normalized ratio of active to total Rap1 is shown (n = 3 independent experiments that are biological replicates). **(C, E)** Hippocampal neurons were transfected with control or knockdown vectors (RNAi, green) for C3G (C) or Rap1 (E) and cultured in NBM or exposed to a hyper-osmotic shock at 1 d.i.v. for 48 h with NBM supplemented with 100 mM sucrose or sorbitol. Neurons were stained at 3 d.i.v. with anti-acetylated tubulin (blue) and anti-Map2 (magenta) antibodies. **(C, D, E, F)** Percentage of unpolarized neurons without an axon (0), polarized neurons with a single axon (Axon), and neurons with supernumerary axons (Spn) is shown for (C, E) (n = 3 independent experiments that are biological replicates). **(G)** Hippocampal neurons were transfected with vectors for control or constitutively active Pip5k1α (αCA, green) and cultured in NBM supplemented with 50 mM sucrose or sorbitol at 1 d.i.v. for 48 h. Neurons were stained at 3 d.i.v. with anti-acetylated tubulin (blue) and anti-Map2 (magenta) antibodies. **(G, H)** Percentage of unpolarized neurons without an axon (0), polarized neurons with a single axon (Axon), and neurons with supernumerary axons (Spn) is shown for (G) (n = 3 independent experiments that are biological replicates). White arrows mark axons. Scale bars: 20 μm. Values are means ± SEM. **(B, D, F, H)**: one-way ANOVA. Source data are available for this figure.

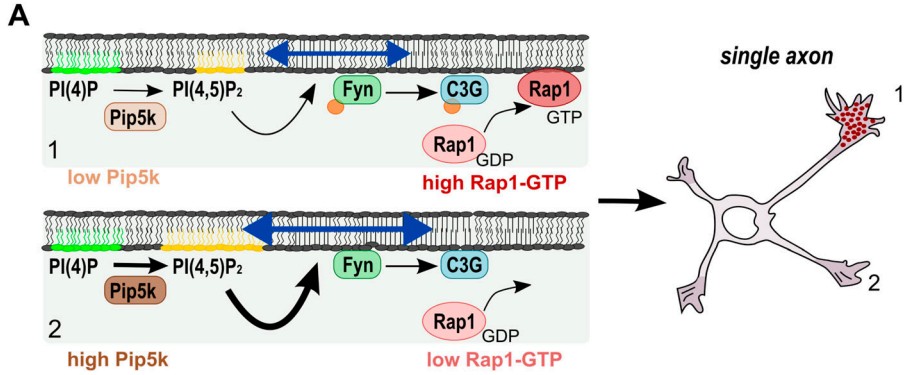

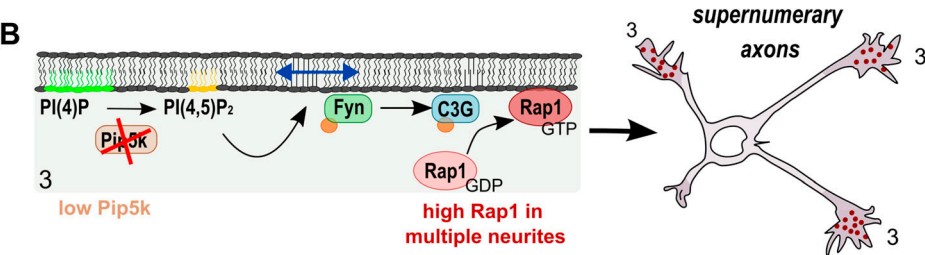

**Figure 7. Regulation of Rap1 by PI(4,5)P2.**
**(A)** In unpolarized neurons, PI(4,5)P2 production by Pip5k1γ restricts the activation of the C3G/Rap1 pathway to a single neurite by modulating the activity of Fyn that phosphorylates and activates the Rap1-GEF C3G and, thereby, Rap1 GTPases. The activation of Rap1 in a single neurite (1) and the repression of its activity in the remaining neurites (2) lead to the formation of a single axon (1). **(B)** Loss of Pip5k1γ strongly reduces the levels of PI(4,5)P2 resulting in the de-repression and activation of the Fyn/C3G/Rap1 pathway in multiple neurites (3), which leads to the formation of supernumerary axons.

# Materials and Methods

**Reagents and Tools table.**

| Reagent/Resource | Source | Identifier |
|---|---|---|
| Experimental Models | | |
| HEK293T | MPI für Hirnforschung (Frankfurt) | N/A |
| Wistar rat | Harlan Winkelmann | N/A |
| Recombinant DNA | | |
| pCAG-EGFP | Matsuda and Cepko (2004) | 11150; Addgene |
| pGEX-4T-2 | Merck Millipore | GE28-9545-50 |
| pSHAG-1 | Paddison et al (2002) | N/A |
| pcDNA6.2-GW/EmGFP-miR | Invitrogen | K493600 |
| pCMV6-Myc-DDK-Pip5k1a | OriGene | MR208688 |
| pCMV6-Myc-DDK-Pip5k1b | OriGene | MR20779 |
| pCMV6-Myc-DDK-Pip5k1c | OriGene | MR209646 |
| pWZL-Neo-Myr-Flag-Pip5k1A | Boehm et al (2007) | 20580; Addgene |
| pWZL-Neo-Myr-Flag-Pip5k1B | Boehm et al (2007) | 20581; Addgene |
| pcDNA6.2-GW/EmGFP-Pip5k1α RNAi | This study | N/A |
| pcDNA6.2-GW/EmGFP-Pip5k1β RNAi | This study | N/A |
| pcDNA6.2-GW/EmGFP-Pip5k1γ RNAi | This study | N/A |
| pcDNA6.2-GW/H2B-mRFP-miR | This study | N/A |
| pcDNA6.2-GW/H2B-mRFP-Pip5k1α RNAi | This study | N/A |
| pcDNA6.2-GW/H2B-mRFP-Pip5k1β RNAi | This study | N/A |
| pcDNA6.2-GW/H2B-mRFP-Pip5k1γ RNAi | This study | N/A |
| pCMV6-Pip5k1α RNAi-res | This study | N/A |

| | | |
|---|---|---|
| pCMV6-Pip5k1β RNAi-res | This study | N/A |
| pCMV6-Pip5k1γ RNAi-res | This study | N/A |
| pGFP-C1-PLCδ1-PH | Stauffer et al (1998) | 21179; Addgene |
| pNES-GFP-C1-PLCδ1-PH | This study | N/A |
| pSHAG-Rap1 shRNAi | Schwamborn & Püschel (2004); Wang et al (2018) | N/A |
| pcDNA6.2-GW/EmGFP-C3G RNAi | This study | N/A |
| pEGFP-C3G | This study | N/A |
| pGEX-GST-RBD | Ren et al (1999) | 15247; Addgene |
| Antibodies | | |
| Mouse anti-acetylated tubulin | Sigma-Aldrich | Cat# T7451; RRID: AB_609894 |
| Rabbit anti-microtubule-associated protein 2 (Map2) | Millipore | Cat# AB5622; RRID: AB_91939 |
| Neuron-specific beta-III tubulin (Tuj1) | R&D SYSTEMS (Bio-Techne) | Cat# MAB1195; RRID: AB_357520 |
| Mouse anti-GFP (GF28R) | Thermo Fisher Scientific | Cat# A5-15256; RRID: AB_10979281 |
| Mouse anti-HA-Tag (6E2) | Cell Signaling Technology | Cat# 2367; RRID: AB_10691311 |
| Mouse anti-myc (9E10) | Sigma-Aldrich | Cat# MABE282; RRID: AB_11204521 |
| Rabbit anti-Pip5k1A | Sigma-Aldrich | Cat# HPA029366; RRID: AB_10602116 |
| Rabbit anti-Pip5k1B | Sigma-Aldrich | Cat# HPA009687; RRID: AB_1855391 |
| Rabbit anti-Pip5k1C | Sigma-Aldrich | Cat# HPA017168; RRID: AB_1855395 |
| Rabbit anti-Rap1 | Millipore | Cat# 07-916; RRID: AB_2177126 |
| Mouse anti-active Rap1 | NewEast Biosciences | Cat# NB-26912; RRID: AB_2629414 |
| Mouse anti-C3G (G-4) | Santa Cruz Biotechnology Inc. | Cat# sc-17840; RRID: AB_628203 |
| Mouse anti-phospho-C3G (G-2) (Tyr514) | Santa Cruz Biotechnology Inc. | Cat# sc-365994; RRID: AB_10918461 |
| Rabbit anti-C3G | Bethyl Laboratories | Cat# A301-965A; RRID: AB_1547894 |
| Rabbit anti-phospho-Src family (Tyr416) | Cell Signaling Technology | Cat# 6943; RRID: AB_10013641 |
| Mouse anti-Fyn (E-3) | Santa Cruz Biotechnology Inc. | Cat# Sc-365913; RRID: AB_10842309 |
| Goat anti-mouse Alexa Fluor 647 | Thermo Fisher Scientific | Cat# A-21235; RRID: AB_2535804 |
| Goat anti-mouse Alexa Fluor 350 (IgG2b) | Thermo Fisher Scientific | Cat# A-21140; RRID: AB_2535777 |
| Goat anti-rabbit Alexa Fluor 488 | Thermo Fisher Scientific | Cat# A-11008; RRID: AB_143165 |
| Goat anti-rabbit Alexa Fluor 594 | Thermo Fisher Scientific | Cat# A-11012; RRID: AB_141359 |
| Goat anti-rabbit Alexa Fluor 647 | Thermo Fisher Scientific | Cat# A-21246; RRID: AB_2535814 |
| Goat anti-rat Alexa Fluor 594 | Thermo Fisher Scientific | Cat# A-11007; RRID: AB_141374 |
| HRP anti-mouse | Dianova | Cat# 115-035-003; RRID: AB_10015289 |

## Experimental model and subject details

All animal protocols were performed in accordance with the guidelines of the North Rhine-Westphalia State Environment Agency (Landesamt für Natur, Umwelt und Verbraucherschutz). Rats were maintained at the animal facility of the Institute for Integrative Cell Biology and Physiology (University of Münster) under standard housing conditions at four to five per cage with a 12-h light/dark cycle (lights on from 07:00 to 19:00 h) at constant temperature (23°C) with continuous access to food and water. Timed-pregnant rats were set up in-house. Pregnant rats were anesthetized by exposure to isoflurane followed by decapitation, and primary cultures were prepared from embryos at embryonic day 18 (E18). During dissection, neurons from all embryos (regardless of sex) were pooled.

## Neuronal cell culture and transfection

Primary hippocampal or cortical neurons were obtained from embryonic day 18 (E18) rat embryos and transfected by calcium phosphate co-precipitation as previously described (Di Meo et al, 2021). In brief, tissues were dissected, collected in HBSS (Invitrogen), and digested in 0.25% trypsin/EDTA (Invitrogen) for 8 min. The digested tissue was mechanically dissociated in DMEM (1X DMEM [Invitrogen], 2 mM glutamine, 50 U/ml gentamycin, and 10% FCS) and passed through a cell strainer (40 μm pore size; Sarstedt). The dissociated neurons were seeded onto poly-L-ornithine–coated (10 μg/ml; Sigma-Aldrich) coverslips at a density of 40,000 cells/cm$^2$. Neurons were allowed to attach in the presence of DMEM for 2 h, which was then replaced by NBM (1x NBM [Invitrogen], 2 mM

glutamine, 50 U/ml gentamycin, and 2% B27 [vol/vol]). For transfection, the culture medium was replaced 3 h after plating by Opti-MEM (Invitrogen) before the DNA mixture was added. After incubation for 45 min at 37°C at 5% $CO_2$, neurons were washed for 15 min with 1 ml of Opti-MEM, which had been pre-incubated at 37°C at 10% $CO_2$, and the conditioned NBM was added back to the cells. Neurons were transfected at 0 d.i.v. and fixed at 3 d.i.v. if not indicated otherwise.

To induce a mild hyper-osmotic shock, the culture medium (NBM) was replaced at the indicated times with NBM supplemented with 50 or 100 mM sucrose or sorbitol. To induce a hypo-osmotic shock, the culture medium was replaced with NBM diluted to 75% or 50% NBM with water for 48 h. Alternatively, neurons were initially cultured in NBM supplemented with 50 mM sucrose or sorbitol, which was replaced with NBM after 1 d in culture. For analysis by immunoprecipitation and Western blot, primary cortical neurons were seeded onto a poly-L-ornithine–coated (10 µg/ml; Sigma-Aldrich) six-well plate at a density of $1.5 \times 10^5$ cells/cm$^2$. Neurons were lysed in RIPA buffer.

## Plasmids

pWZL-Neo-Myr-Flag-Pip5k1A (Pip5k1Aca, plasmid #20580; Addgene) and pWZL-Neo-Myr-Flag-Pip5k1B (Pip5k1βCA, plasmid #20581; Addgene) were a gift from William Hahn and Jean Zhao, and pGFP-C1-PLCD1-PH was a gift from Tobias Meyer (plasmid #21179; Addgene). Loss-of-function experiments were performed with miRNAs generated using the BLOCK-iT Pol II miRNA Expression Vector Kit (Invitrogen) (Table S1). Target sequences were cloned into the pcDNA6.2-GW/EmGFP-miR expression plasmid following the manufacturer's instructions. The efficiency and specificity of the knockdown were verified by Western blot, immunofluorescence staining, and rescue experiments (Figs S1B and C–E and S2G and H). To generate pcDNA6.2-GW/H2B-mRFP-miR, H2B-mRFP was excised from pLV-RFP with DraI/SalI and inserted into pcDNA6.2-GW/EmGFP to replace the EmGFP sequence. Single point mutations were generated by site-directed mutagenesis using Phusion High-Fidelity DNA Polymerase (Thermo Fisher Scientific). The sequences of the oligonucleotides (Metabion) used are displayed in Table S1. Pip5k1α, Pip5k1β, and Pip5k1γ RNAi-resistant constructs were generated from pMyc-DDK-Pip5k1a (#MR208688; OriGene), pMyc-DDK-Pip5k1b (#MR20779; OriGene), and pMyc-DDK-Pip5k1c (#MR209646; OriGene), respectively. To express C3G (Rapgef1), the complete mouse C3G coding sequence (accession number BC129881) was cloned in the pEGFP-C2 expression vector. The PI(4,5)P$_2$ sensor pGFP-C1-PLCδ1-PH was modified by the addition of a nuclear export signal (NES) as described previously (Simon-Areces et al, 2013) because the expression of PLCδ1-PH was detrimental for neuronal health. The pNES-GFP-C1-PLCδ1-PH was generated through the insertion of the NES sequence into pGFP-C1-PLCδ1-PH (Table S1).

## Immunofluorescence staining

Immunofluorescence staining of cultured neurons was carried out as previously described (Di Meo et al, 2021). The specificity of the anti-Pip5k1α, anti-Pip5k1α, and anti-Pip5k1α antibodies was verified by a loss of the signal after knockdown (Fig S1C and D). Neuronal cultures were fixed with 4% PFA and 4% sucrose for 15 min at 37°C and quenched in 50 mM ammonium chloride at RT for 10 min. After several washes with PBS, the cells were permeabilized with 0.1%

Triton X-100 for 3 min and treated with blocking solution (2% normal goat serum, 2% BSA, and 0.2% fish gelatin in PBS) for 1 h at RT. Fixed cells were incubated overnight with primary antibodies at 4°C. Cells were washed with PBS, incubated with the appropriate Alexa Fluor–conjugated secondary antibody (Thermo Fisher Scientific) at RT for 2 h, and mounted using Mowiol (Sigma-Aldrich). Antibodies were diluted in 10% blocking solution.

## GST pull-down, immunoprecipitation, and Western blot

For pull-down assays to analyze the amount of active Rap1, the GST-RBD (Ras/Rap-binding domain) fusion protein was expressed in *E. coli* BL21 cells as described previously (Katoh et al, 2015) and immobilized on Glutathione Sepharose 4B beads (GE Healthcare) for 1 h at 4°C. After three washes (20 mM Tris–HCl, pH 7.4, 25 mM NaCl, 0.1 mM DTT, and complete protease inhibitor), they were incubated with cell lysates at 4°C for 4 h. The beads were washed three times (50 mM Tris–HCl, pH 7.4, 150 mM NaCl, 1 mM DTT, 1.5 mM MgCl$_2$, 5 mM EDTA, 10% [vol/vol] glycerol, 0.1% [vol/vol] Triton X-100, and complete protease inhibitor), and bound proteins were eluted with 50 µl of 2x SDS sample buffer and analyzed by Western blot. For the immunoprecipitation of endogenous proteins, cell lysates were incubated with the phospho-SFK-Tyr416 antibody (1:100) at 4°C for 4 h. Protein G agarose beads (Merck) were added and incubated at 4°C for 1 h. After three washes (50 mM Tris–HCl, pH 7.4, 150 mM NaCl, 1 mM DTT, 1.5 mM MgCl$_2$, 4 mM EDTA, 10% [vol/vol] glycerol, 1% [vol/vol]Triton X-100, and complete protease inhibitor), bound proteins were eluted with 50 µl of 2x SDS sample buffer and analyzed by Western blot.

Proteins were separated by SDS–polyacrylamide gel electrophoresis and transferred to nitrocellulose membranes. Nonspecific binding was blocked at RT with 5% non-fat dry milk or 3% fish gelatin (for detection of phosphorylated proteins; Sigma-Aldrich) in Tris-buffered saline and 0.1% Tween-20 (TBST) for 1 h. The membrane was incubated overnight with primary antibodies diluted in blocking solution at 4°C. After several washes with TBST, membranes were incubated with HRP-coupled secondary antibodies (Jackson ImmunoResearch Labs) for 2 h at RT. Peroxidase activity was visualized by the enhanced chemiluminescence detection system (Interchim) using the ChemiDocTM MP imaging system (Bio-Rad).

## Image acquisition and analysis

Images were acquired on a Zeiss LSM 800 Airyscan laser scanning confocal microscope with 405-, 488-, 568-, and 647-nm laser lines and processed using Zeiss ZEN 2.3 (blue edition) software (Carl Zeiss MicroImaging). Neuronal morphology was analyzed in fixed neurons using a Plan-Apochromat 40x/1.3 Oil DIC M27 objective.

## Quantification and statistical analysis

### Quantification

Neuronal polarity was analyzed at 3 d.i.v. as described previously (Di Meo et al, 2021; Ravindran & Püschel, 2023). Neurons with neurites that were positive for Map2 but negative for acetylated tubulin were categorized as unpolarized, and neurons with a single long and

acetylated tubulin–positive neurite as normally polarized (Witte et al, 2008). Neurons with several long neurites showing accumulation of acetylated tubulin were counted as neurons with supernumerary axons. At least 150 cells per condition were counted for each experiment. The polarization index was calculated as the fluorescence intensity (A.U.) of the brightest neurite divided by the average fluorescence intensity (A.U.) of the other neurites of the same neuron.

### Statistical analysis

All plotted values are expressed as means ± SEM. All experiments were performed independently at least three times. Specific numbers can be found in the figure legends. If not otherwise indicated, comparisons between two groups were performed using an unpaired *t* test, and comparisons of more than two groups were done using one-way ANOVA followed by Tukey's test with GraphPad Prism. For all analyses performed, significance was defined as ns: $P > 0.05$; *$P \leq 0.05$; **$P \leq 0.01$; ***$P \leq 0.001$; and ****$P \leq 0.0001$. When not indicated, the comparison did not show a significant difference.

## Data Availability

Data sets generated from the current study are available from the corresponding author upon reasonable request.

## Supplementary Information

## Acknowledgements

We thank Maria Wenning, Ina Kowsky, and Verena Stegemann for technical assistance. This work was supported by the Deutsche Forschungsgemeinschaft (DFG) through the Cells-in-Motion Cluster of Excellence (EXC 1003-CiM), grants PU 102 14-1 and SFB 1348.

### Author Contributions

D Di Meo: conceptualization, data curation, formal analysis, investigation, visualization, methodology, and writing—original draft, review, and editing.
T Kundu: data curation, formal analysis, investigation, visualization, methodology, and writing—review and editing.
P Ravindran: data curation, formal analysis, investigation, visualization, methodology, and writing—review and editing.
B Shah: formal analysis, investigation, and methodology.
AW Püschel: conceptualization, resources, data curation, supervision, funding acquisition, validation, visualization, project administration, and writing—review and editing.

### Conflict of Interest Statement

The authors declare that they have no conflict of interest.

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
