## [Reviewer comments · Life Science Alliance]

Life Science Alliance

Pip5k1 γ regulates axon formation by limiting Rap1 activity

Danila Di Meo, Trisha Kundu, Priyadarshini Ravindran, Bhavin Shah, and Andreas Puschel

DOI: <https://doi.org/10.26508/lsa.202302383>

Corresponding author(s): *Andreas Puschel, University of Münster*

Review Timeline:

Submission Date:	2023-09-19
Editorial Decision:	2023-10-19
Revision Received:	2024-01-08
Editorial Decision:	2024-02-22
Revision Received:	2024-02-26
Accepted:	2024-02-26

Scientific Editor: *Eric Sawey, PhD*

Transaction Report:

October 19, 2023

Re: Life Science Alliance manuscript #LSA-2023-02383

Prof. Andreas W. Puschel
University of Münster
Institut für Integrative Zellbiologie und Physiologie
Schlossplatz 5
Münster 48149
Germany

Dear Dr. Puschel,

Thank you for submitting your manuscript entitled "Pip5k1 γ regulates axon formation by limiting Rap1 activity" to Life Science Alliance. The manuscript was assessed by expert reviewers, whose comments are appended to this letter. We invite you to submit a revised manuscript addressing the Reviewer comments.

Thank you for this interesting contribution to Life Science Alliance. We are looking forward to receiving your revised manuscript.

Sincerely,

B. MANUSCRIPT ORGANIZATION AND FORMATTING:

Reviewer #1 (Comments to the Authors (Required)):

The definition of axonal fate has been under a lot of scrutiny. However nowadays it is not clear how the axonal fate could be distinguished from axon elongation. In other words, how one the neurite is selected to eventually grow as an axon versus the mechanism sustaining axonal growth once the fate is established. In addition, it is not clear, once the axon is growing, what is the mechanism that precludes the other neurites (future dendrites) from growing.

The authors claim that Pip5k (lipid-modifying enzyme) is the limiting factor precluding neurite elongation in minor neurites once the axon is growing. Thus, low levels of Pip5k in the growing axon allow the activation of Rap1 which promotes axon extension.

I have some concerns regarding this manuscript:

1. The author failed to cite several manuscripts dealing with axon selection and axonal growth. For instance, this is not the first study trying to explain how minor neurites or future dendrites are not growing meanwhile the axon is extending. There are manuscripts from M.M. Poo and Kaibushi labs that directly try to address this issue and are not cited in the current manuscript. Furthermore, if the authors claim that Pip5k is behind axon selection, they also need to consider the manuscripts that try to explain axon selection based on the intracellular organization of the cytoplasm.
2. The authors failed to clarify whether Pip5k regulates axon fate or if only mediates axon extension. The title suggests that they tackle axonal fate; however, there is no data showing the role of Pip5k in stage 2 neurons or before (when the axon is still not visually evident). For instance, they do not show if before axon formation Pip5k is enriched in the neurites that will not grow during axon extension. Therefore, it is not clear if low levels of Pip5k in the growing axon are only promoting axon extension or if this kinase is instrumental in axon selection.
3. It would be beneficial for this study to show the participation of Pip5k during axon formation in situ.
4. Why are the authors not using an axonal marker? Certainly, acetylated tubulin is enriched in the axon; still, it could be better to use an axonal marker.
5. Some pictures are not clear given that blue is used. Why not use red?

Reviewer #2 (Comments to the Authors (Required)):

A key event in establishing neuronal morphology happens in early stages of neuron differentiation where one of the neurites begins to extend rapidly as the axon, while the remaining ones do not grow until they differentiate into dendrites. In this paper, Di Meo et al. observed that knockdowns of three different isoforms of Pip5k1 result in the formation of supernumerary axons in cultured hippocampal and cortical neurons at early differentiation stages (d.i.v 0-3). This phenotype is specific for each isoform and is coupled to a reduction of the levels of the membrane phospholipid PI(4,5)P2. Rescue experiments with various constructs revealed that loss of neuronal polarity due to gamma isoform knockdown can be restored by expressing constitutively active (CA), membrane-targeted Pip5k1 alpha and beta. The authors then measured the levels of the Fyn/Rap1/C3G signaling pathway involved in axon formation, which all increase in response to sustained or transient inhibition of PIP5k1 gamma. However, only the sustained inhibition of Pip5k1 gamma can maintain increased C3G/Rap1 activity. The role of C3G and Rap1 in Pip5k1-dependent regulation of axonal formation was further confirmed by knockdown of C3G or Rap1, which suppressed the formation of supernumerary axons after inhibition of Pip5k1 gamma. Finally, the authors explored the connection between membrane tension, Pip5k1 gamma activation and axonal formation by applying various protocols of hyper and hypo-osmotic shock to cultured neurons. Both transient and sustained hyper-osmotic shock triggered formation of supernumerary axons, while sustained hypo-osmotic shock impaired axon formation. Notably, knockdown of C3G or Rap1 as well as expression of constitutively active Pip5k1 prevent the formation of supernumerary axons induced by hyper-osmotic shock, establishing a link

between membrane tension, Pip5k1 activity and PI(4,5)P2 production.

In this study, the authors propose a Pip5k1y-dependent mechanism regulating axonal formation through PI(4,5)P2 production and modulation of C3G and Rap1 activity. High Pip5k1 activity and PI(4,5)P2 production restrict Rap1/C3G activity in neurites, while low Pip5k1 activity and reduced PI(4,5)P2 levels might change membrane tension, leading to de-repression of the Rap1/C3G pathway, neurite outgrowth and consequent axon formation. Overall, I consider that the conclusions of this study are well substantiated and the proposed molecular mechanism is thoroughly explained and fits with the experimental data. I believe that only a few points need to be addressed to bring this study to completion:

- It would be important to know whether the supernumerary axons in various conditions (Pip5k1 knockdown and inhibition, hyper-osmotic shock) have similar length between themselves as well as compared to control axons.
- In Figure 5, hyper-osmotic shock was applied for either 48h or 5h. However, hypo-osmotic shock was only applied for 48h in two different contexts. Is there a reason for this discrepancy and, if not, could the authors also test the effect of hypo-osmotic shock for 5h?
- The authors suggest that the reversion of induction of supernumerary axons by Rap1/C3G knockdown or expression of CA-Pip5k1 in context of hyper-osmotic shock suggests a link between PI(4,5)P2 production, membrane tension and axon formation. However, there are no measurements of PI(4,5)P2 levels in different contexts of osmotic shock. If PI(4,5)P2 is indeed a key effector in regulation of supernumerary axons by restricting Rap1 and C3G, then one would expect its levels to decrease in hyperosmotic shock as this triggers formation of supernumerary axons and increased Rap1 activity. Similarly, expressing CA-Pip5k1a would increase the levels of PIP2 and explain the reversion of the phenotype. To confirm this, PIP2 levels could be measured in different contexts of osmotic shock using the same PIP2 sensor as in Figure 1.
- Last, there are a couple of instances where the literature cited is incorrect or missing. Line 80: the cited references only apply to the second half of that sentence, please rephrase as also the evidence that membrane tension affects SLKs is only based on osmotic shocks and is thus tenuous. Line 339: the De Belly et al paper is not showing that membrane tension acts only locally. Lines 340-342: Both the statement and references are wrong, can the authors please be more precise?
- The arguments in lines 323 and 324 should be inverted and further substantiated as suggested above.
- What are the red arrows in Figure 3? In the legend only the yellow ones are explained.

Reviewer #1

We thank the reviewer for the helpful comments. We agree that it is probably not possible to differentiate between axon specification and extension and have modified the manuscript accordingly.

1. The author failed to cite several manuscripts dealing with axon selection and axonal growth. For instance, this is not the first study trying to explain how minor neurites or future dendrites are not growing meanwhile the axon is extending. There are manuscripts from M.M. Poo and Kaibushi labs that directly try to address this issue and are not cited in the current manuscript. Furthermore, if the authors claim that Pip5k is behind axon selection, they also need to consider the manuscripts that try to explain axon selection based on the intracellular organization of the cytoplasm.

1) We included additional references and clarifications as suggested by the reviewer (p. 3). These references also cover the intracellular mechanisms more extensively.

The formation of a single axon is thought to be directed by a combination of positive and negative feedback signals. Our results show that Pip5k activity acts as an inhibitory signal to prevent minor neurites from becoming an axon. We do not suggest that a decrease in Pip5k activity also initiates the formation of the axon during neuronal polarization (see also response to point 2).

2. The authors failed to clarify whether Pip5k regulates axon fate or if only mediates axon extension. The title suggests that they tackle axonal fate; however, there is no data showing the role of Pip5k in stage 2 neurons or before (when the axon is still not visually evident). For instance, they do not show if before axon formation Pip5k is enriched in the neurites that will not grow during axon extension. Therefore, it is not clear if low levels of Pip5k in the growing axon are only promoting axon extension or if this kinase is instrumental in axon selection.

2) The reviewer asked to clarify whether Pip5ks are involved in axon specification or extension. We agree that it is "not clear how the axonal fate could be distinguished from axon elongation" because a rapid growth is essential for the formation of an axon. Since a neurite has to extend beyond a minimal length to become an axon, it may not be possible to separate axon extension and specification (Goslin and Banker, 1989; Toriyama et al., 2010; Yamamoto et al., 2012). For this reason, we mostly used "axon formation" as a generic term instead of "axon specification" and changed the sentences (pp. 2, 4 and 9) that still referred to "axon specification".

Our results indicate that inhibition of PI(4,5)P2 production disrupts an inhibitory signal that prevents minor neurites from becoming an axon. We do not propose that a decrease in Pip5k activity is also the signal that initiates the formation of the axon during neuronal polarization. A decrease in PI(4,5)P2 levels may contribute to axon formation that is initiated by a positive, growth-promoting signal but this remains to be investigated in future experiments.

We analyzed the distribution of Pip5ks in unpolarized neurons as suggested (new Supplementary Figures S1A and S1B). Immunofluorescence staining does not show a differential distribution of Pip5ks in unpolarized neurons (Fig. S1A and S1B) or at a stage when the axon is rapidly growing (Fig. S1C). The function of Pip5ks does not depend on the enrichment in specific neurites because their activity is regulated by phosphatidic acid and GTPases.

3. It would be beneficial for this study to show the participation of Pip5k during axon formation in situ.

3) We agree that a conditional knockout or in utero electroporation to investigate the function of Pip5k in the embryonic brain would be interesting. However, our study focuses on the cellular mechanism that restrain axon growth. We think that additional in vivo experiments would go beyond the scope of our study.

4. Why are the authors not using an axonal marker? Certainly, acetylated tubulin is enriched in the axon; still, it could be better to use an axonal marker.

4) The reviewer asked why we used acetylated tubulin as an axonal marker. Acetylated tubulin was shown by Dr. Bradke and colleagues to be an excellent and specific axonal marker (Witte et al., 2008). In our hands, it is a more robust and reliable marker at this stage than markers like the Tau-1 and SMI-312 antibodies. These antibodies recognize specific dephospho- (Tau-1) and hyper-phosphorylated (SMI-312) epitopes, respectively. Both antibodies, therefore, indicate not only axonal properties but also the activity of multiple kinases, which may be affected by phospholipids. The selection of acetylated tubulin as axonal marker avoids that a possible modulation of kinase activity affects the analysis of axon formation.

5. Some pictures are not clear given that blue is used. Why not use red?

5) We did not encode immunofluorescence signals with a combination of red and green to accommodate all viewers as recommended by guidelines for image presentation.

Reviewer #2

We thank the reviewer for the positive comments and addressed the remaining open questions as detailed below.

• It would be important to know whether the supernumerary axons in various conditions (PIP5k1 knockdown and inhibition, hyper-osmotic shock) have similar length between themselves as well as compared to control axons.

1) We included a quantification of axonal length as Supplementary Figures S2F and S4A. The length of supernumerary axons extended after suppression of Pip5ks was not significantly different from controls (Fig. S2F). Axons extended after a hyper-osmotic shock were shorter than in controls (Fig. S4A), possibly because the treatment modulates pathways in addition to Pip5ks (Diz-Muñoz et al., 2016, Loh et al., 2019).

• In Figure 5, hyper-osmotic shock was applied for either 48h or 5h. However, hypo-osmotic shock was only applied for 48h in two different contexts. Is there a reason for this discrepancy and, if not, could the authors also test the effect of hypo-osmotic shock for 5h?

2) We included the suggested experiment in Fig. 5E. A hypo-osmotic shock for 5 hours had the same effect as one for 48 hours. The effect was less pronounced, probably because during a shorter incubation fewer neurons initiate their transition to a polarized morphology when axon formation can be affected by the treatment.

• The authors suggest that the reversion of induction of supernumerary axons by Rap1/C3G knockdown or expression of CA-Pip5k1 in context of hyper-osmotic shock suggests a link between PI(4,5)P2 production, membrane tension and axon formation. However, there are no measurements of PI(4,5)P2 levels in different contexts of osmotic shock. If PI(4,5)P2 is indeed a key effector in regulation of supernumerary axons by restricting Rap1 and C3G, then one would expect its levels to decrease in hyperosmotic shock as this triggers formation of supernumerary axons and increased Rap1 activity. Similarly, expressing CA-Pip5k1a would increase the levels of PIP2 and explain the reversion of the phenotype. To confirm this, PIP2 levels could be measured in different contexts of osmotic shock using the same PIP2 sensor as in Figure 1.

3) We tested if an osmotic shock affects the level of PI(4,5)P2 as suggested (new Supplementary Figures S4B and S4C). These experiments show a significant reduction of the signal for PI(4,5)P2 after a hyperosmotic shock. We also tested the effect of a hypo-osmotic shock but neurons expressing the PI(4,5)P2 sensor were extremely sensitive to the treatment and most transfected neurons did not survive in contrast to controls or a hyper-osmotic shock.

• Last, there are a couple of instances where the literature cited is incorrect or missing. Line 80: the cited references only apply to the second half of that sentence, please rephrase as also the evidence that membrane tension affects SLKs is only based on osmotic shocks and is thus tenuous. Line 339: the De Belly et al paper is not showing that membrane tension

acts only locally. Lines 340-342: Both the statement and references are wrong, can the authors please be more precise?

4) We rephrased the sentence in line 80 as suggested. We had cited De Belly et al. (line 339) because this publication highlights previous results that membrane tension acts mainly locally (Shi et al., 2018) while it reports that tension can be propagated over longer distances when the actin cortex is engaged. We modified lines 339 to 342 to better explain this point.

- *The arguments in lines 323 and 324 should be inverted and further substantiated as suggested above.*

5) We modified the sentence (lines 323 and 324) as suggested.

- *What are the red arrows in Figure 3? In the legend only the yellow ones are explained.*

6) We thank the reviewer for spotting this omission and included the information in the legend for Fig. 3.

February 22, 2024

RE: Life Science Alliance Manuscript #LSA-2023-02383R

Prof. Andreas W. Puschel
University of Münster
Institut für Integrative Zellbiologie und Physiologie
Schlossplatz 5
Münster 48149
Germany

Dear Dr. Puschel,

Thank you for submitting your revised manuscript entitled "Pip5k1 γ regulates axon formation by limiting Rap1 activity". We would be happy to publish your paper in Life Science Alliance pending final revisions necessary to meet our formatting guidelines.

- please address the Reviewer's remaining comments
- please be sure that the authorship listing and order is correct
- please add the Twitter handle of your host institute/organization as well as your own or/and one of the authors in our system
- there is only one supplementary table provided, and it is labeled as Table S2, please correct accordingly

A. FINAL FILES:

B. MANUSCRIPT ORGANIZATION AND FORMATTING:

Sincerely,

Reviewer #2 (Comments to the Authors (Required)):

A key event in establishing neuronal morphology happens in early stages of neuron differentiation where one of the neurites begins to extend rapidly as the axon, while the remaining ones do not grow until they differentiate into dendrites. In this paper, Di Meo et al. observed that knockdowns of three different isoforms of Pip5k1 result in the formation of supernumerary axons in cultured hippocampal and cortical neurons at early differentiation stages (d.i.v 0-3). This phenotype is specific for each isoform and is coupled to a reduction of the levels of the membrane phospholipid PI(4,5)P2. Rescue experiments with various constructs revealed that loss of neuronal polarity due to gamma isoform knockdown can be restored by expressing constitutively active (CA), membrane-targeted Pip5k1alpha and beta. The authors then measured the levels of the Fyn/Rap1/C3G signaling pathway involved in axon formation, which all increase in response to sustained or transient inhibition of Pip5k1 gamma. However, only the sustained inhibition of Pip5k1 gamma can maintain increased C3G/Rap1 activity. The role of C3G and Rap1 in Pip5k1-dependent regulation of axonal formation was further confirmed by knockdown of C3G or Rap1, which suppressed the formation of supernumerary axons after inhibition of Pip5k1 gamma. Finally, the authors explored the connection between membrane tension, Pip5k1 gamma activation and axonal formation by applying various protocols of hyper and hypo-osmotic shock to cultured neurons. Both transient and sustained hyper-osmotic shock triggered formation of supernumerary axons, while sustained hypo-osmotic shock impaired axon formation. Notably, knockdown of C3G or Rap1 as well as expression of constitutively active Pip5k1 prevent the formation of supernumerary axons induced by hyper-osmotic shock, establishing a link between membrane tension, Pip5k1 activity and PI(4,5)P2 production.

In this study, the authors propose a Pip5k1y-dependent mechanism regulating axonal formation through PI(4,5)P2 production and modulation of C3G and Rap1 activity. High Pip5k1 activity and PI(4,5)P2 production restrict Rap1/C3G activity in neurites, while low Pip5k1 activity and reduced PI(4,5)P2 levels might change membrane tension, leading to de-repression of the Rap1/C3G pathway, neurite outgrowth and consequent axon formation. Overall, I consider that the conclusions of this study are well substantiated and the proposed molecular mechanism is thoroughly explained and fits with the experimental data. The authors addressed our comments and suggestions from the first submission round, further strengthening the message of the study. Thus, I strongly support publication in LSA.

But if possible I would still ask that two minor remarks are addressed:

- The authors have quantified axonal length in various experimental conditions (Pip5k1 knockdown and inhibition, hyper-osmotic shock) and found that Pip5k1 knockdown does not affect the length of supernumerary axons compared to controls (Fig. S2F). Could the authors speculate why this is the case? If there were a possible physical limit for Pip5k1y-dependent neurite extension one would expect a difference in length... Moreover, axons extended in hyper-osmotic shock are shorter than in controls (Fig. S4A); according to the authors, hyper-osmotic shock modulates other "pathways in addition to Pip5k1y" (lines 279-281). Since both the cited references in this sentence do not correspond to studies performed in neurons, it would be important to suggest which alternative pathways could be involved in axonal "shrinking".
- The authors have now successfully demonstrated that PI(4,5)P2 levels significantly decrease after an hyperosmotic shock (Fig. S4B and S4C), further reinforcing the link between PI(4,5)P2 production, membrane tension, and axon formation. While we acknowledge that similar measures were challenging for hypo-osmotic shock due to the survival of primary neurons, is there a reason for not measuring the PI(4,5)P2 levels after hyper-osmotic shock when CA-Pip5k1a is expressed?

Reviewer #2

We thank reviewer 2 for the positive comments.

1) The reviewer asked why the suppression of Pip5ks does not reduce the length of supernumerary axons. Our results suggest that Pip5k activity restricts the extension of neurites. One would not necessarily expect that axons are shorter when this restriction is removed and there is no direct evidence for a physical limit of neurites length as suggested by the reviewer.

The reviewer also asked to name pathways that are responsible for "axonal shrinking" after a hyperosmotic shock. However, supernumerary axons are shorter because they extend less than the untreated control, not because they shrink. We can only refer to pathways known from non-neuronal cells (see cited publications) since additional pathways that are modulated by an osmotic shock during axon formation remain to be analyzed.

2) We could not analyze PI(4,5)P₂ production for all conditions (i.e. after expression of constitutively active Pip5k1a combined with a hyper-osmotic shock) because the expression of the PI(4,5)P₂ sensor can compromise the viability of neurons. This was especially evident for some expression vectors when they were combined with a stressful treatment, even when expression levels were carefully titrated.

February 26, 2024

RE: Life Science Alliance Manuscript #LSA-2023-02383RR

Prof. Andreas W. Puschel
University of Münster
Institut für Integrative Zellbiologie und Physiologie
Schlossplatz 5
Münster 48149
Germany

Dear Dr. Puschel,

Thank you for submitting your Research Article entitled "Pip5k1 γ regulates axon formation by limiting Rap1 activity". It is a pleasure to let you know that your manuscript is now accepted for publication in Life Science Alliance. Congratulations on this interesting work.

DISTRIBUTION OF MATERIALS:

Again, congratulations on a very nice paper. I hope you found the review process to be constructive and are pleased with how the manuscript was handled editorially. We look forward to future exciting submissions from your lab.

Sincerely,
